# A New Approach to Estimate from Monitored Demand Data the Limit of the Coverage of Electricity Demand through Photovoltaics in Large Electricity Grids

**DOI:** 10.3390/s20164390

**Published:** 2020-08-06

**Authors:** Francisco Baena, Francisco José Muñoz-Rodriguez, Pedro Gómez Vidal, Gabino Almonacid

**Affiliations:** 1Department of Electronic and Automatic Engineering, University of Jaén, Campus Lagunillas, 23071 Jaén, Spain; 2Centro de Estudios Avanzados en Energía y Medio Ambiente CEAEMA, Universidad de Jaén, 23071 Jaén, Spain; fjmunoz@ujaen.es (F.J.M.-R.); pvidal@ujaen.es (P.G.V.); galmona@ujaen.es (G.A.)

**Keywords:** coverage electricity demand, matching demand-photovoltaic, photovoltaics, renewable integration, demand-side, large electricity grid

## Abstract

In a traditional large electricity grid without storage, there is a limit to the maximum photovoltaic energy that can be consumed as the demand and generation may not match, either in magnitude or in time. This paper aims to provide a new method to estimate the limit of the coverage of electricity demand by photovoltaics in large electricity grids. This new method eliminates the random and the periodic variability over time as it is based either on the load duration curve for demand and the output duration curve for PV generation. We will assume there is no energy storage or inter-network exchanges. Moreover, conditions for the best scenario for photovoltaics are provided in order to estimate the upper limit: photovoltaic overgeneration is not considered and a complete system flexibility is assumed. The knowledge of this limit will manage to provide not only a reference for the planning of the energy sector but also to analyze the viability of the integration of future photovoltaic projects in the electrical system. In order to illustrate the method, several large electricity grids have been analysed in order to determine the aforementioned limit. Values between 19.3% and 29.9% have been obtained.

## 1. Introduction

### 1.1. Maximum Coverage of the Demand by Photovoltaics. Definition

Photovoltaic energy is one of the generation sources known as variable energy resources (VERs) [1,2]. The main drawback of these energy sources is that they cannot completely match the demand curve. From a balancing perspective, the intermittent nature of these renewable energy sources are creating challenges in the electricity sector that previously were not of concern [3]. Moreover, there may be shortcomings with the availability and surplus of electricity depending on the irradiance and demand at a given time. Likewise, this availability is determined by the intrinsic intermittence of the solar resource: the periodic variability due to the variation throughout the year of the relative positions between the Earth and the Sun, and the random variability introduced by the atmosphere, mainly due to clouds and the variable activity of the Sun in much lower levels [4].

The daily output power curve of a photovoltaic (PV) power generation plant, in the absence of storage, follows the incident solar radiation. In this way, it provides, on a clear day, a maximum on solar noon and canceling at dawn. Since electricity cannot be stored on a large scale with simple and/or cost-competitive technologies, the generation of electrical power must occur at the same time this power is consumed. This implies that the electrical system, as a whole, has to be always ready to supply the electricity that is being demanded [5].

Furthermore, the demand for electricity in large electricity grids, with millions of consumers, is characterized by its daily load curve which remains above zero throughout the day with a minimum during the dawn and one or more peaks during the day and/or at first hours of the night. The behavior of this load curve generally depends on the climatology and the uses and customs of the population.

Both curves, PV generation and demand, cannot completely match as each one depends on different factors. Furthermore, the matching between these two curves has also the added difficulty that they both change independently throughout the year. Therefore, the matching between them, conditioned by the shape and magnitude of each curve, determines the amount of photovoltaic power that can be injected into the power grids and be absorbed by the demand (E_matchDPV_). In this way, it can be expressed as [6]:(1)$$E_{\mathrm{matchDPV}}=E_{\mathrm{Dem}}\cdot C_{T}\cdot C_{\mathrm{Mag}}$$
where C_T_ is a coefficient due to the matching between the PV generation time and the time in which demand occurs. This coefficient is given by the relation between the maximum time of PV generation over a period of time and the total duration of demand in the same period (its maximum value is 1). For photovoltaics without storage in large conventional networks it will be around 0.5. C_Mag_ is a coefficient that reflects the matching between the magnitudes of the PV generation and demand. It is obtained by the average photovoltaic power harvested during the generation time divided by the average demand power during the same interval. Its maximum value is 1, although without PV overcapacity this coefficient will be significantly lower than 1.

Without going into other considerations and only taking into account the behavior of the demand and photovoltaic generation curves, in a traditional power grid, it can be said that the matching between them will never be 100%, without considering any kind of electricity storage. This fact will imply that there will be a limit on the maximum coverage of the demand by photovoltaics (CD_PVmax_). This parameter, or average penetration, is the proportion of electric energy which is being supplied from photovoltaic plants. The coverage of demand is defined by the following equation [5]: (2)$${\mathrm{CD}}_{\mathrm{PVmax}}=\frac{E_{{\mathrm{matchDPV}}_{\max}}\;}{E_{\mathrm{Dem}}}$$

The limit of the coverage of electricity demand for a given electricity generation technology is an indicator of the potential of this technology. Therefore, the limit of the CD_PV_ (CD_PVmax_) is an indicator of the ability of photovoltaic solar energy to cover the electricity demand for a given grid. Solar PV has a limitation when considering the amount of energy that can be absorbed by a large conventional electrical grid. This is due to the matching between the demand curve and the photovoltaic generation curves. Currently, in most networks, the installed photovoltaic power is far from reaching this limit. However, the accumulated PV power is continuously increasing and getting closer to this limit. The CD_PV_ limit may provide, given the aforementioned boundary conditions, the maximum photovoltaic power that a given grid can absorb. In this sense, this limit can be very useful in energy planning of large electricity grids as it provides certainty to future projects of photovoltaic installations about the capacity of the demand to absorb the photovoltaic energy that can be generated.

It must be highlighted that this limit refers only to the matching between electricity demand and PV generation, regardless of the limits imposed by the energy resource [7], the necessary material for photovoltaic installations [8,9], transport in the grid [10] or other restrictions [11]. Therefore, in order to estimate this limit of the CD_PV_, it is necessary to study both the behavior of the electricity demand for a specific network and the one corresponding to the PV generation that is injected into it.

Currently, although the level of CD_PV_ is low, there are very significant growth estimations. In the countries with the highest coverage it was slightly higher than 8% while in EU and the world as an average it reached 5% and 3%, respectively, in 2019 [12]. However, this coverage will increase significantly as an annual growth of the new photovoltaic power installed of 9% is expected for the period 2020–2024 [13]. If this growth is kept in the next decades, there will be many countries with a considerable increase in CD_PV_ where PVs may become one of the main sources of electricity generation Nowadays, in most networks, the installed photovoltaic power is still far from reaching CD_PVmax_. However, the accumulated PV power is continuously increasing and getting closer to this limit. The CD_PV_ limit may provide, given the aforementioned boundary conditions, the maximum photovoltaic power that a given network can absorb without storage and inter-network exchanges.

### 1.2. Methods to Estimate the Upper Limit of the CDPV in a Large Electricity Grid. State of the Art

Several researchers have explored the integration of photovoltaic power generation into conventional power grids, although only a few have tried to estimate the technical limit of the penetration of PV with respect to the demand. In the different published papers on this subject it is observed that the most generalized method to determine the limit of the CD_PV_ is based on comparing the electricity demand data for a given network and for a particular year with the PV generation data obtained indirectly from the solar radiation data in the network area for the same year [14,15,16,17].

In all these papers, the PV energy injected into the network is estimated by computer simulation, assuming determined characteristics of the PV system connected to the network, such as the type of tracking, the overall performance of PV installations, etc. In this way, the electricity demand curve for the whole year is compared with the simulated annual PV generation curve for the same year. Moreover, the value of the instantaneous PV power is scaled for the whole year to different values of the demand power (either referred to the maximum annual power or annual average power), considering whether or not there is overcapacity of PV generation, i.e., if the instantaneous PV power exceeds the demand at a given time and in what magnitude it exceeds. Also, it is considered whether there is energy storage to absorb this PV generation excess, exchanges between networks or if it is directly lost. All this leads to different scenarios that estimate different limits of the CD_PV_ that prevent any kind of comparison. The block diagram of Figure 1 succinctly shows the aforementioned methods.

This method, which is used in the different referenced papers, estimates the limit of CD_PV_ through Equation (3) [5]:(3)$${\mathrm{CD}}_{\mathrm{PV}}=\frac{E_{\mathrm{PV}}}{E_{\mathrm{Dem}}}=\frac{\sum_{n=1}^{365}\sum_{i=1}^{24}P_{n,i}^{\mathrm{PV}}∆t}{\sum_{n=1}^{365}\sum_{i=1}^{24}P_{n,i}^{\mathrm{Dem}}∆t}$$
where P^PV^_ni_ y P^Dem^_ni_ define the average powers of each sample *i* taken at a given recording interval Δt, for each n day of the year, for the photovoltaic generation and the demand, respectively. 

This equation is valid for the PV power as long as the latter is equal to or less than the demand power. In this sense, the excess of PV generation or PV overgeneration that takes place is not considered. The PV overgeneration can be defined as the generation of PV electricity above the one needed to meet the demand. This can be lost or stored. Whereas, PV overcapacity is defined as the availability of installed PV power above the need for PV generation. The latter may be conditioned by a limitation in the output power.

The aforementioned papers provide a range of values for the limit of the coverage of the demand that depends mainly on the flexibility of the network, the degree of PV overcapacity, the level of storage and, in some cases, the cost of kWh expressed as a function of the previous variables. A limit of the CD_PV_ is given for an extreme scenario of 100% flexibility, without storage and without PV over-generation. All of them consider a conventional demand profile (that of for the considered year) for the analysed networks [14,16,17,18]. The data corresponding to the networks of the state of Texas are from the year 2000, in the state of New York they are from the year 2005, in the metropolitan area of Zurich they are from October 2007 to September 2008 and the network of Switzerland is from the year 2005. The limit obtained in these papers is 22% (considering a 65% flexibility for the electricity grid regarding photovoltaics), 14.83%, 17% (with a 1% of PV overcapacity) and 24%, respectively. The last case is not comparable to the other three works, since, in addition to the comments at the beginning of this section, it considers a flexibility of 75% and a single ideal day of PV generation, on 20 June 2007, which is extrapolated for the whole year.

The values proposed by these authors are not absolute limits of the CD_PV_, even in this extreme scenario, since in all cases they are determined for a specific year and a specific network, both for PV generation and demand, which do not guarantee that in that year and in that network the best conditions are given to obtain the maximum matching between the PV generation and the electricity demand. In addition, none of these studies are carried out in different years or networks in order to make a comparison of results.

Furthermore, the main drawback is that the aforementioned method depends strongly on the variation of both demand and PV generation over time. For example, if the maximum annual power of PV generation coincides on Sunday, it will provide a different limit if it falls on Wednesday. Moreover, different limits will be obtained if this maximum value coincides with a heat wave or it takes place on an average temperature day.

This paper aims to present a new method to estimate the maximum limit in the coverage of the demand by photovoltaics imposed by the coupling between the demand and the photovoltaic generation in a large and classic electricity grid. The knowledge of this limit may be of special interest for the energy planning of a territory and for the management of the electric network operators. The method here developed is based on the load duration curve (LDC) for demand and the output duration curve (ODC_PV_) for PV generation. This new method has the advantage of eliminating the random and the periodic variability over time of the instantaneous PV power and, to a lesser extent, of the electricity demand. This fact will be shown in the next sections. In this sense, it makes possible to estimate the upper limit in the CD_PV_ in a simple and intuitive way, as the aforementioned curves are stable for a given network and for different years [6]. It should be noted that this limit is estimated considering a determined scenario: without electrical storage, without photovoltaic overgeneration, i.e., all the photovoltaic electricity generated must be absorbed by the demand considering total flexibility of the electrical system, and using the conventional ODC_PV_ and LDC profiles of each network.

In order to achieve the aforementioned objective, the paper will be structured as follows. In Section 2 the new method based on LDC and ODC_PV_ curves is developed. Next, in Section 3 the results obtained when applying this method to several large electricity grids are presented. Finally, the most relevant conclusions are drawn in Section 4.

## 2. Materials and Methods

### 2.1. Estimation of the Upper Limit of the CD_PV_ in a Large Electricity Grid: Proposed Method

A new method for the estimation of the upper limit of the CD_PV_ in large electricity grids with conventional demand profiles will be developed, Figure 2. The latter is based on the use of the load duration curve (LDC) for demand and the output duration curve for PV generation (ODC_PV_).

These curves provide a transformation of the data used in the aforementioned methods. In the latter, the power is plotted as a function of the time. However, both LDC or ODC_PV_ are curves rearranged hour-by-hour in chronological order based on power magnitude [19], i.e., they show how long a certain power is maintained in a decreasing monotone way. These curves can be daily, weekly, monthly or annual. In this paper, only annual curves will be considered (8760 h/year).

In this way, this transformation eliminates the temporal variability of the data and makes it possible to obtain the limit of the coverage of the demand avoiding the noise that incorporates this variability, noise that for this limit is indifferent or, at least, of little relevance. However, this noise makes it difficult to obtain the limit of CD_PV_ when considering other methods.

These curves have the advantage of maintaining their shape for different years [20]. The ODC_PV_ depends on the type of solar tracking of the PV plants, the dispersion between plants of the same large-scale PV system, the PV generation overcapacity and the annual radiation profile incident on the photovoltaic field. Meanwhile, the LDC of a large power grid depends mainly on the electricity consumption habits of the network population and the climatology. Radiation and climatology undergo variations if observed at a small scale (e.g., a day, a week or a month), but they are stable if observed at a larger scale of time such as a year. The other variables remain stable over the years.

Furthermore, both curves eliminate random and periodic variability of power with respect to time, since they do not represent the power with respect to chronological time but with respect to the number of hours that this power is maintained at a certain level, obtaining a decreasing monotonous representation [21,22]. This issue, together with a determined scenario, manages to develop a simple and effective method for the estimation of the limit the PV’s can reach in the coverage of the demand according to the profiles of the LDC and the ODC_PV_.

In order to estimate the upper limit of the CD_PV_, it will be defined the best scenario for PV’s will be defined: the annual demand profile of a large electricity grid will be considered in its normalized LDC form stable over the years and with 100% electrical system flexibility to give maximum priority to PV’s (i.e., this would imply subordinating all other sources participants in the grid energy mix to the PV’s). Moreover, neither excess of PV generation nor any kind of energy storage will be considered. To complete this scenario, the following conditions are defined to obtain the maximum matching between the demand and the PV generation:(1)The maximum annual output peak power of the PV generation matches, in magnitude and in time, with the maximum annual power of the demand.(2)For every day of the year, the instantaneous demand power will always be equal to or greater than the instantaneous power of the PV generation.

These conditions are clearly theoretical and are only of interest in providing the best scenario, since any other conditions will determine a lower CD_PV_ limit. However, it must be said that the first condition can be easily achieved, or at least, got close to it, in electricity grids of warm zones, where the maximum annual demand power of demand is reached in summer and near noon. An example of this is case of the ERCOT network in Texas [14].

With respect to the second condition, it is necessary to emphasize the sharp decline in the demand that takes place at weekends and during holidays. However, the monotonic decreasing PV generation curve (ODC_PV_) has two characteristics that facilitate compliance with this point. The first one is that its duration is around half the duration of the demand curve (LDC), while the second one is that the power decay slope is higher in the ODC_PV_ than in the LDC. Moreover, it must be highlighted that this problem diminishes as we move away from the summer solstice.

Despite the two issues mentioned above, the sharp decrease in demand at weekends and during holidays would mean that in these days the second condition set to achieve the maximum CD_PV_ is not met. However, in most of the networks studied, the PV generation decreases in a proportion equal to or greater than the demand in a number of days similar to that of weekends and holidays (about 110 days), both circumstances do not coincide in time. In order to take into account this second condition it can be assumed that the weather conditions throughout the year reduce the PV generation at weekends and during public holidays without changing the annual balance. This implies assuming a hypothetical translation of the PV generation in time, without modifying its magnitude and therefore the ODC_PV_, as shown in Figure 3.

It must be highlighted that in these types of curves (LDC and ODC_PV_) the time axis is not ordered following the arrow of time (i.e., past towards the future) but attending to the value of the power of each temporary sample. In these curves, the average hourly power is classified from its maximum value to its minimum value, regardless the time in which it occurred [23].

Figure 4 shows an example of LDC, where the area under the curve represents the energy consumed if both axes are in real magnitudes (W and h). In this way, for the demand energy (E_Dem_), the area under the LDC may be given as:(4)$$E_{Dem}=\overset{L}{\underset{i=1}{\sum }}P_{Di}∆t$$
where *L* is the number of annual hours, *P_Di_* is the average hourly power demanded and Δ*t* is the recording interval (1 h).

In contrast, when the power axis is represented in a normalized form, referred to the maximum annual power consumed (P_max-a_), the area under the normalized curve represents the total number of hours at the maximum annual power (t_p-a_) that would have been needed to match the energy demanded in the considered year. If both axes are normalized in parts per unit (pu) the area under the curve has no dimensions and it will provide the annual load factor (LF) of the system.

The area under the curve can be expressed as equivalent rectangular surfaces, as can be seen in Figure 4. This approach manages to obtain the value of the area as the product of the base by height. So that the annual electric energy demanded can be expressed by Equation (5), using either the maximum annual power together with t_p-a_ or the average annual power (P_avg-a_) and the maximum annual time (t_max-a_) for demand.
(5)$$E_{Dem}=P_{max-a}\cdot t_{p-a}=P_{avg-a}\cdot t_{max-a}$$

Regarding PV generation, the area below the ODC_PV_ curve, Figure 5, is given by Equation (6) which estimates the photovoltaic energy generated throughout the year:(6)$$E_{PV}=\overset{L}{\underset{i=1}{\sum }}P_{Gi}∆$$
where *L* is the number of annual hours, *P_Gi_* corresponds to the average hourly power generated and Δ*t* is the time interval (1 h).

When analysing the ODC_PV_ curve, the following considerations should be taken into account, Figure 5.
The range of the abscissa axis when considering power values is about half the one-year hours for PV systems without storage (4380 h in a non-leap year). This is because, without considering any kind of energy storage, a PV plant can generate electricity, on average, about 12 h every day of the year.ODC_PV_ is delimited by the following parameters: P_max-a_ (maximum annual hourly AC power generated) and t_g-a_ (annual PV generation time). This last one represents the total number of hours per year that the PV plant has been supplying power to the grid and it is around half the annual maximum time.Two additional parameters can be defined on ODC_PV_: the average power of photovoltaic generation for the time t_g-a_, P_avg-a(tg-a)_, and the average power, for the maximum annual time (t_max-a_ = 8760 h for a non-leap year), is given by P_avg-a_.

As was done with LDC, the same area under the ODC_PV_ can be represented by equivalent rectangular areas using the previously defined parameters of power and time.

In this sense, the area under the ODC_PV_, can be expressed in different magnitudes:⚬If the two axes are in real values (W-h) the area under the ODC_PV_ represents the output energy of the PV system given in Wh/year.⚬If the power axis is normalized with respect to the maximum annual power (pu) and the time axis is maintained in real values (h), the area under the ODC_PV_ corresponds to the annual final yield given in hours/year, that is t_vm_. The latter can be defined as the number of equivalent annual hours operating at maximum annual AC power measured at the output of the system to generate the annual energy produced. This parameter, t_vm_, can be also defined as the ratio between the annual photovoltaic energy injected into the network (E_PV_) and Pmax-a, as the maximum effective aggregate power of the set of plants that constitute the PV system. It must be highlighted that t_vm_ do not correspond to the system final yield as the latter is calculated by dividing the net energy output of the entire system in AC (this value does match E_PV_) per rated kW (DC) of installed power [24].⚬If both axes are normalized (pu), the area under ODC_PV_ has no dimensions and it will provide the dimensionless ratio of electrical PV energy output over a given period of time to the maximum possible electrical energy output over that period, i.e., the capacity factor (CF) of the PV system.

Therefore, these rectangular areas represent the CF, the annual final yield (in large power grid) given in hours or the photovoltaic energy injected into the network, depending on whether the axes are defined in pu or in real values, respectively. In this sense, the output energy of the PV system can be expressed by the equation:(7)$$E_{PV}=t_{vm}P_{max-a}=t_{g-a}P_{avg\left(tg-a\right)}=t_{max-a}P_{avg-a}$$

Therefore, the theoretical limit that can be reached by the PV generation is determined by the relation between the PV energy injected into the grid and the energy demanded by the same network. Up to to this point, there are no changes to the method used in the cited literature [14,16,17]. However, as already mentioned, the new method here developed is based on the ODC_PV_ and LDC curves since the area under them represents the corresponding energies of PV generation and demand, respectively. In this way, the coverage of the demand by solar photovoltaics, CD_PV_, may be given by the following expression [6,20]:(8)$$CD_{PV}=\frac{LDC_{area}\cap ODC_{PVarea}}{LDC_{area}}$$
where the numerator of Equation (8) represents the useful PV consumed energy due to demand. This ratio implies that if the whole area under the ODC_PV_ curve is included in the area under the LDC, the limit of the CD_PV_ can be expressed by Equation (9), if conditions 1 and 2 of the best scenario are fulfilled:(9)$$ODC_{PV-area}\in LDC_{area}\;⟹\;\;CD_{PV}=\frac{ODC_{PV-area}}{LDC_{area}}$$

For LDC and ODC_PV_ curves, these areas can be expressed as a function of their respective power summations as a function of time (Equations (4) and (6), respectively) or by means of equivalent rectangular areas (Equations (5) and (7)). These equivalent rectangular areas for demand and PV generation are shown in Figure 6 where the associated parameters for equivalent rectangular area are also given.

Using Equations (5) and (7) the limit of the CD_PV_ can be estimated by the ratio of the different parameters associated with the equivalent rectangular areas, as given in the equation below:(10)$$CD_{PV}=\frac{E_{PV}}{E_{Dem}}=\frac{t_{vm}}{t_{p-a}}=\frac{P_{avg-a}^{PV}}{P_{avg-a}^{Dem}}=\frac{t_{g-a}\cdot P_{avg\left(tg-a\right)}^{PV}}{t_{max-a}\cdot P_{avg-a}^{Dem}}$$

Comparing the last term of Equation (10) with Equation (1), coefficients C_T_ and C_Mag_ may be obtained. Therefore, the CD_PV_ may be expressed as the product of these two coefficients.
(11)$$C_{T}=\frac{t_{g-a}}{t_{max-a}}$$
(12)$$C_{Mag}=\frac{P_{avg\left(tg-a\right)}^{PV}}{P_{avg-a}^{Dem}}$$
(13)$$CD_{PV}=C_{T}\cdot C_{Mag}$$

Any of the relations shown in Equations (10) and (13) can be used to determine the CD_PV_ limit. The choice of one or the other will depend only on the ease in obtaining the corresponding parameters.

As can be seen, the main advantage of this method is that it manages to obtain an upper limit in the coverage of the demand with PVs regardless both the grid and the year. This limit only depends on the profile of the LDC and ODC_PV_ curves, which are stable for different years and very similar between grids of similar latitudes and similar consumption habits [25].

This maximum limit will be reached if the scenario set out at the beginning of this section is fulfilled. In the next section, and in order to illustrate the proposed method, the latter will be applied to several large scale electricity grids from different countries in order to determine the limit of the coverage of the demand by means of PVs.

### 2.2. Data 

The developed method will be applied to several large scale electricity grids of different countries considering data of both photovoltaic and electricity demand for the same period of time: Amprion, Tennet, Transnet and 50 Hertz from Germany, Elia from Belgium and REE-Tenerife from Spain. Table 1 shows general data of each grid, the covered area and population. The electricity demand data of a given electricity grid are provided by their corresponding Transmission System Operators (TSO) websites of each electricity grid, both for photovoltaic generation [26,27,28,29,30,31] and for the demand [32,33,34,35,36,37]. TSOs must make these data public in accordance with Regulation (EU) No 543/2013 [38]. The data to be downloaded correspond to the average power, the recording interval and the time and date of each sample.

As a necessary preliminary step, the data preprocessing should be highlighted in order to provide the appropriate format for the analysis.

The source data present recording intervals of 10, 15, 30 and 60 min, depending on the TSO. In this way, the recording interval has been normalized to one hour. An interval of 60 min has been chosen because it is the least common multiple of all the available intervals. To transfer all the data from the different grids to this common sampling interval, the following equation has been used:(14)$$P_{i}=\frac{1}{N}\overset{\begin{array}{c} i=24\\ j=N \end{array}}{\underset{\begin{array}{c} j=1\\ i=1 \end{array}}{\sum }}P_{ij}\;\mathrm{para}\;N=\frac{60\;min}{∆t}$$
where *P_i_*: Average power in the normalized recording interval, *P_ij_*: average power in the original recording interval, Δ*t*: original recording interval given in minutes and *N*: number of originals recording intervals for each normalized recording interval.

In addition, due to the significant difference in size between the different grids, it is also neccesary to normalize the peak power or anual maximun power as it varies from 8 MW to 100 GW. In this sense, the power has been normalized in parts per unit (pu) [5] to the annual máximum power using the following expression:(15)$$P_{i\left(pu\right)}=\frac{P_{i}}{P_{max-a}}$$
where *P_i(pu)_*: Normalized power in pu, *P_i_*: Average power in the normalized recording interval (1 h) and *P_max-a_*: Annual maximun power.

Moreover, it should be noted that there are two issues regarding the source data that must be corrected:
Incoherent data and lack of data: both the absence of data and incoherent data. Estimation techniques have been used such as the quantitative method of time series called “Trend projection” [39] in order to correct these anomalies.The time change: throughout the European Community [40] the official time will be advanced one hour on the last Sunday of March and will be delayed one hour on the last Sunday of October. In North America it is similar, one hour is advanced on the second Sunday of March and is delayed on the first Sunday of November. 

Figure 7 and Figure 8 shows annual data, once processed, provided by the TSO (German 50 Hertz), corresponding to photovoltaic generation and demand, respectively.

In order to apply the method previously described in Section 2, these data are transformed into their corresponding ODC_PV_ and LDC. In order to estimate the limit of CD_PV_, the ratio between t_vm_ and t_p-a_ shown in Equation (10) will be considered. However, any other quotient in this equation can be also used.

## 3. Results and Discussion

Figure 9 shows the ODC_PV_ (blue shaded area) and LDC (red shading area) curves of these six large electricity grids, as well as the estimated value of the CD_PV_ for each one. The axes in all graphs are normalized to the annual maximum value, in both power and time. It must be highlighted that the first four networks (a–d) represent the average year obtained between 2012 and 2103, both for photovoltaic and for demand, while the in the last two ones (e–f) the data only corresponds to 2013, the only year available.

Figure 9 is completed with the data shown in Table 2 which includes t_vm_, t_p-a_ and CD_PV_ values for the six large electricity grids studied. In those grids where it has been possible to study more than one year, the average value of the studied years has been also included. As can be seen in this Table, CD_PV_ varies between 19.3% and 19.5% for Elia (Belgium) and 50 Hertz (Germany), respectively, and the maximum value, 29%, is reached by REE-Tenerife (Spain). All these values correspond to 2013 and they have been determined under the conditions given in Section 2.

In the case of REE-Tenerife, it can be seen that both the t_vm_ and t_p-a_ are the highest ones of the networks considered. Moreover, and as it has been aforementioned this electricity grid also provides the maximum CD_PV_ of all the networks studied. This is clearly due to t_vm_, which is approximately 50% higher than the other central European electricity grids (this is due to its climatology and its geographical location which is very close to the Tropic of Cancer) while its t_p-a_ is only slightly higher. Whereas, the minimum values obtained in Elia and 50 Hertz networks are due to both a decrease in their t_vm_ and a high level of t_p-a_.

Through this method, if LCD and OCD_PV_ curves are known, the aforementioned boundary conditions are set and using Equation (10), the CD_PVMAX_ in a given network can be determined only knowing four parameters of the electricity grid: annual energy and maximum annual power, both for demand and for PV generation. This fact may be of interest when analysing networks that have the same LCD and ODC_PV_ profiles. If these curves are not available, the method shown in Figure 2 will have to be applied, i.e., firstly it should be obtained LCD and ODC_PV_ curves for that network should be obtained. Moreover, this method has the advantage of obtaining an upper limit in the CD_PV_ in a simple and intuitive way, which is stable for a given electricity grid and for different years. The knowledge of this limit may be of special interest for the energy planning of a territory and for the management of the electricity grids operators.

It must be highlighted that the values obtained by this method may differ from those that would be obtained by the methods presented in Section 1 as it eliminates the random and the periodic variability of the power both demand and generation.

## 4. Conclusions

In this paper a new method has been developed to estimate the limit of the CD_PV_ in large electricity grids. Solar PV has a limitation when considering the amount of energy that can be absorbed by a large conventional electrical network. This is due to the matching between the demand curve and the photovoltaic generation curves. Currently, in most networks, the installed photovoltaic power is far from reaching this limit. However, the accumulated PV power is continuously increasing and may get closer to this limit in the next decades. The CD_PVMAX_ may provide, given the aforementioned boundary conditions, the maximum photovoltaic power that a given network can absorb if neither storage nor inter-network exchanges are considered. 

The method has the advantage of eliminating the random and the periodic variability over time of the PV power and, to a lesser extent, the demand power. This is achieved through the ODC_PV_ and the LDC curves which, together with the best scenario conditions to maximize the CD_PV_, make it possible to obtain the theoretical limit which the PV generation can achieve for a given electricity grid. This limit depends both on the profiles of LDC and ODC_PV_, both stable over time for an electricity grid. 

The CD_PVMAX_ constitutes a theoretical limit that shows the potential of this technology and the ability of PVs to cover the electricity demand for a given grid. Values beyond this limit are not possible if no electrical storage and photovoltaic overgeneration are considered. For this reason, to know this limit can be very useful in energy planning of large electricity grids as it provides certainty to future projects of photovoltaic installations about the capacity of the demand to absorb the photovoltaic energy that can be generated. It can be very useful in energy planning of large electricity grids as it provides certainty to future projects of photovoltaic installations about the capacity of the demand to absorb the photovoltaic energy that can be generated. It must be stated that the limit developed in the manuscript provides the potential of PVs alone when facing the demand.

In order to illustrate the method, the latter has been applied to several electricity grids. For the analysed grids, this limit in the CD_PV_ moves between 19.3% and 29.9% of the annual energy demand in the corresponding electricity grid. It should be noted that these values are determined without storage, without PV generation overcapacity and with the conventional ODC_PV_ and LDC profiles of each network. This limit is not universal but depends on the LDC and the ODC_PV_ of each grid.

In addition, this limit can be increased by means of improvements in PV generation, such as the incorporation of the geographic dispersion between PV plants, solar tracking, PV overcapacity and the storage of photovoltaic generation surplus. Furthermore, by managing the electricity demand, this limit can be also increased, e.g.: if the peak of daily demand shifts towards solar noon and the peak of annual demand towards the summer solstice, the valley of daily demand during the night or increases demand on weekends and holidays. Future works should be developed in this sense in order to know the effect of the aforementioned factors when determining the limit on the coverage of the demand by photovoltaics.

The practical limit of the CD_PV_ that can be reached in conditions of network security, guarantee of supply and competitive costs depends on multiple factors that are beyond the scope of this work.

## Figures and Tables

**Figure 1 sensors-20-04390-f001:**
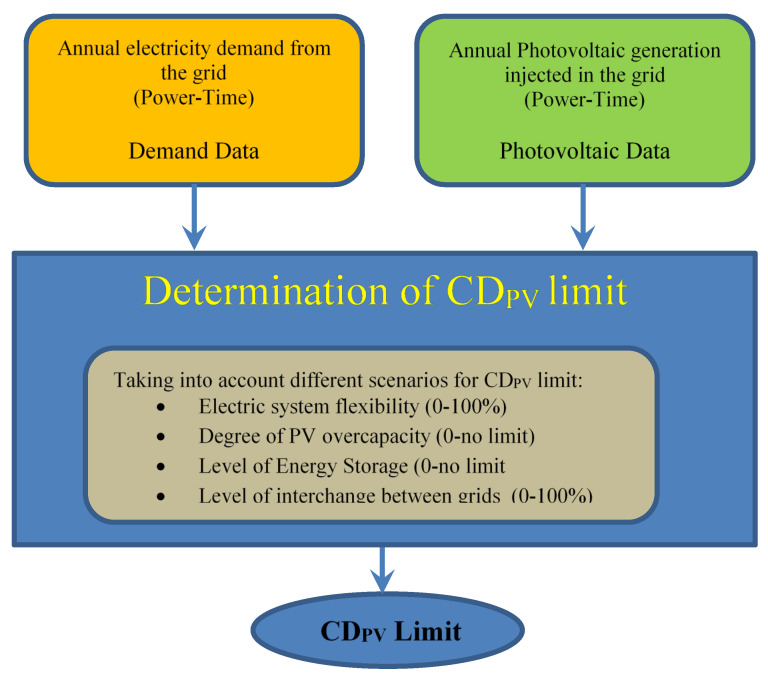
Methods used in the literature to estimate the limit of the coverage of electricity demand by means of PV’s. As it can be seen, different scenarios can be considered.

**Figure 2 sensors-20-04390-f002:**
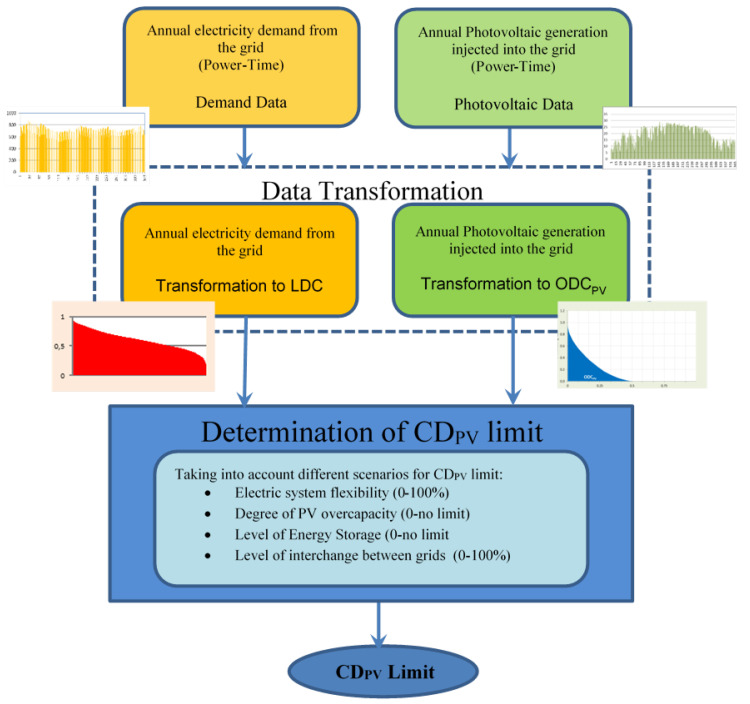
Proposed method for the estimation of CD_PV_ in a large electricity grids.

**Figure 3 sensors-20-04390-f003:**
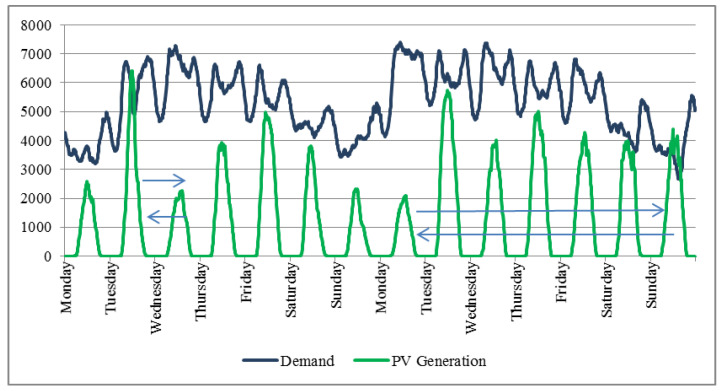
Theoretical displacement of the PV generation curves in order to achieve the best scenario (2nd condition) for CD_PVMAX_. Real data corresponding to Transnet network (9–22 April 2012).

**Figure 4 sensors-20-04390-f004:**
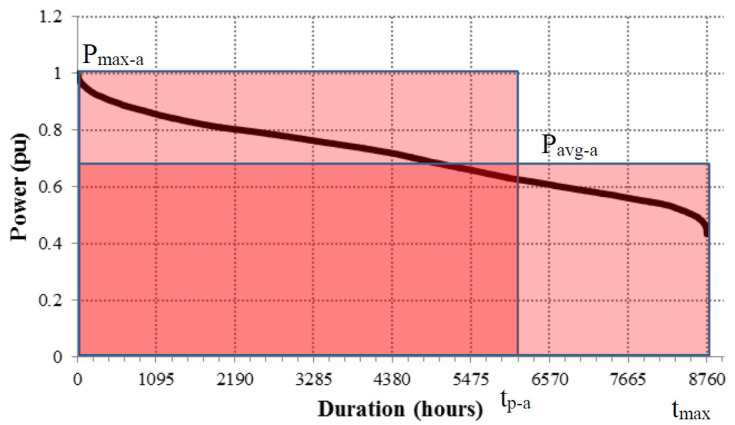
LDC and equivalent rectangular areas.

**Figure 5 sensors-20-04390-f005:**
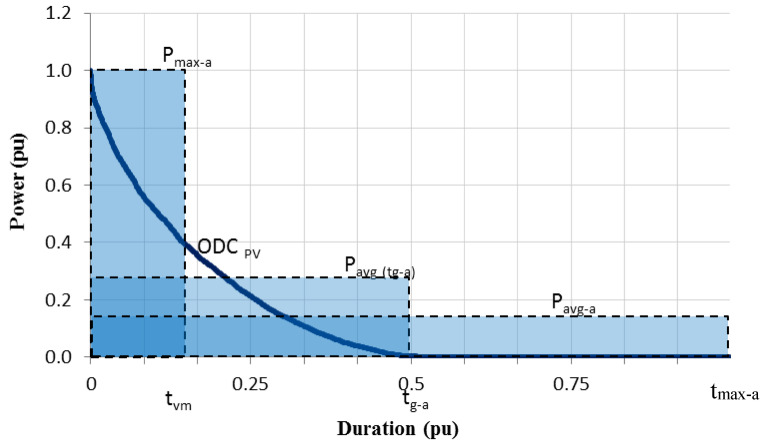
ODC_PV_ and their equivalent rectangular areas.

**Figure 6 sensors-20-04390-f006:**
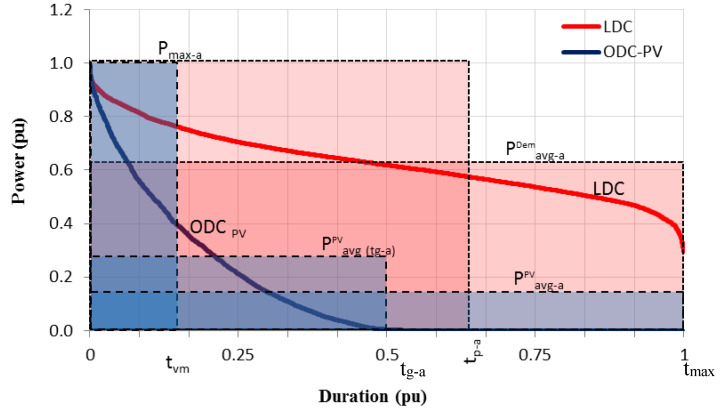
Equivalent rectangular areas for the PV generation (blue) and the demand (red).

**Figure 7 sensors-20-04390-f007:**
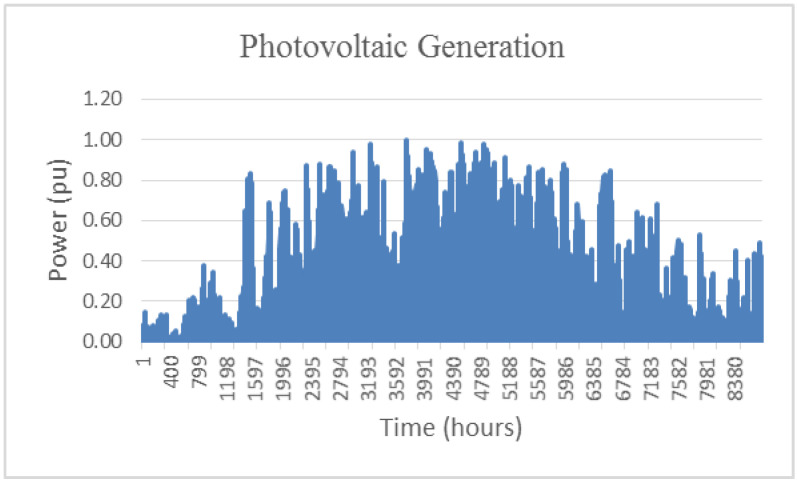
PV generation. Data corresponding to Germany- 50 Hertz (2013)**.**

**Figure 8 sensors-20-04390-f008:**
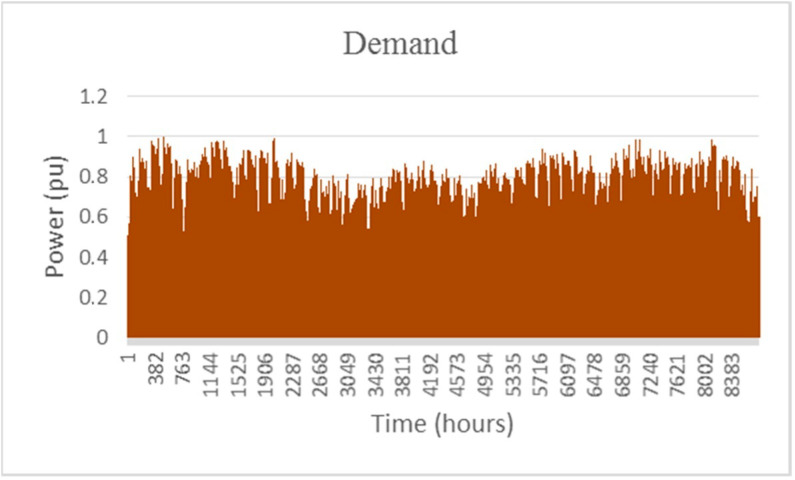
Demand data corresponding to Germany- 50 Hertz (2013)**.**

**Figure 9 sensors-20-04390-f009:**
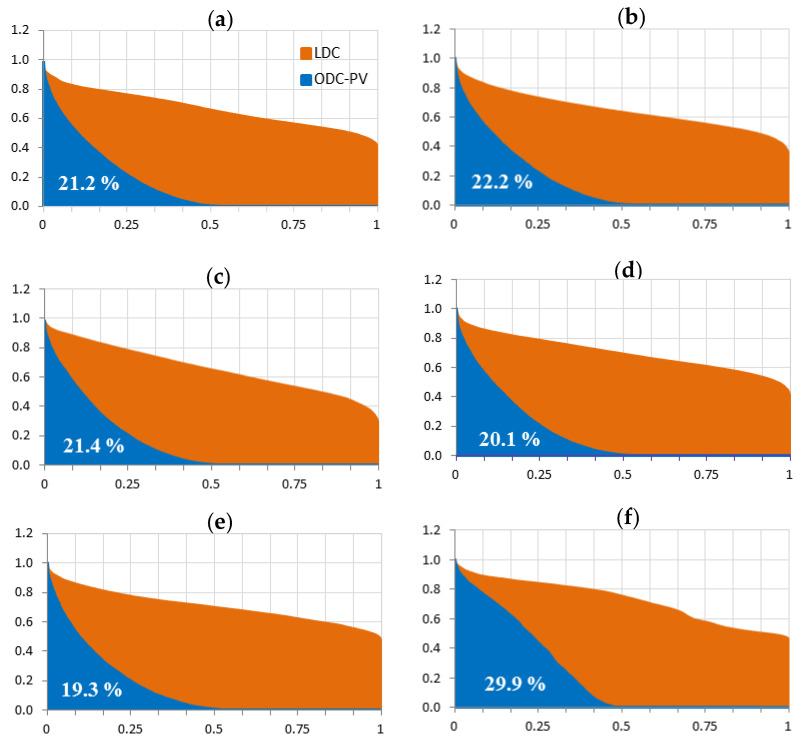
Limit of the CDPV for six european large electricity grids: (**a**) Amprion (Germany), (**b**) Tennet (Germany), (**c**) Transnet (Germany), (**d**) 50 Hertz (Germany), (**e**) Elia (Belgium) and (**f**) REE-Tenerife (Spain, Canary Islands).

**Table 1 sensors-20-04390-t001:** Analyzed electricity grids. General data.

Electricity Grid	Country	Area (km^2^)	Population Attended (10^6^ Inhabitants)
Amprion	Germany	73,100	27
Transnet	Germany	34,600	11
Tennet	Germany	138,780	20
50-Hertz	Germany	109,360	18
Elia	Belgium	30,528	10
REE-Tenerife	Spain	2034	0.9

**Table 2 sensors-20-04390-t002:** CD_PV_ estimation for several European large electricity grids.

Electricity Grid	Year	t_vm_ (hours)	t_p-a_ (hours)	CD_PV_ (%)
Amprion	2012	1240	5630	22
	2013	1217	5972	20.4
	Average	1228	5801	21.2
Tennet	2012	1257	5774	21.8
	2013	1243	5481	22.7
	Average	1250	5623	22.2
Transnet	2012	1202	5332	22.5
	2013	1245	6101	20.4
	Average	1223	5717	21.4
50 Hertz	2012	1289	6106	20.8
	2013	1178	6043	19.5
	Average	1223	6075	20.1
Elia	2013	1185	6134	19.3
REE-Tenerife	2013	1877	6282	29.9

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
