# Peer review of "A New Approach to Estimate from Monitored Demand Data the Limit of the Coverage of Electricity Demand through Photovoltaics in Large Electricity Grids"

_sensors, 2020, doi:10.3390/s20164390_

Round 1

Reviewer 1 Report

The article presents in an. Language is excessively redudant, with long description of simple concepts as if the readers are totally unaware of the topic or as if the authors created something totally new. Referencing is not great. The claimed 'generality' of the method is not fully that: assumptions are still pretty strong, such as contemporaneity of peak PV generation and peak load as well as compatibility of the moments of largest generation. So, yes, you identified the maximum theoretical coverage, but what's the significance if that is not feasibile? Also, the assumption of no storage and no exchange, how is that realistic?

I am sorry to say that the work must be deeply revised.

Author Response

The authors would like to thank you for your revision. We kindly appreciate your comments that have helped to improve the manuscript and to make the text clearer.
Point 1: The article presents in an. Language is excessively redudant, with long description of simple concepts as if the readers are totally unaware of the topic or as if the authors created something totally new.
Response 1: We have modified the manuscript. As you suggest, a deep review has been made and figures 1 and 2 together with the associated text, which may be redundant, have been removed. Moreover, it has been revised by a native speaker.
Poin 2: Referencing is not great.
Response 2: It is very difficult to find references related with this issue. However, we have managed to find more references which have been included in the manuscript.
[3] Miklós Gyalai-Korpos et al. The Role of Electricity Balancing and Storage: Developing Input Parameters for the European. Calculator for Concept Modeling. Sustainability 2020, 12, 811.
[6] F. Baena Villodres. Influence of Photovoltaic Electricity on the Energy Mix. Theoretical limit in coverage for electricity demand by Solar Photovoltaic Energy. Ph.Tesis. University of Jaén. 2017.
[12] IEA PVPS. Snapshot of global photovoltaic Markets 2020. Report IEA PVPS T1-37:2020. April 2020.
[13] Solar Power Europe. Global Market Outlook For Solar Power / 2020-2024. June 2020.
Poin 3: The claimed 'generality' of the method is not fully that: assumptions are still pretty strong, such as contemporaneity of peak PV generation and peak load as well as compatibility of the moments of largest generation.
Response 3: Sorry for the misunderstanding. The method here developed which manages to determine the limit in CDPV may be applied to any large electricity grid, here lies its generality. However, it must be considered, as you say, very strong assumptions to obtain this upper limit, but it is the only way to achieve this goal. These boundary conditions try to define the scenario where there may be only PV to face the demand as it is stated in lines 212-225
“In order to estimate the upper limit of the CDPV, it will be defined the best scenario for PV´s will be defined: the annual demand profile of a large electricity grid will be considered in its normalized LDC form stable over the years and with 100% electrical system flexibility to give maximum priority to PV´s (i.e. this would imply subordinating all other sources participants in the grid energy mix to the PV´s). Moreover, neither excess of PV generation nor any kind of energy storage will be considered. To complete this scenario, the following conditions are defined to obtain the maximum matching between the demand and the PV generation:
1. The maximum annual output peak power of the PV generation matches, in magnitude and in time, with the maximum annual power of the demand.
2. For every day of the year, the instantaneous demand power will always be equal to or greater than the instantaneous power of the PV generation.
These conditions are clearly theoretical and are only of interest in providing the best scenario, since any other conditions will determine a lower CDPV limit.”
It must be emphasized that knowledge of the limitations of a technology is of great importance to make a realistic and effective use of this technology.
Moreover, the paragraph “The method here developed manages to determine the limit in the CDPV where only four parameters of the electricity grid are needed: annual energy and maximum annual power, both for demand and for PV generation.” is replaced by “Through this method, if LCD and OCDPV curves are known, the aforementioned boundary conditions are set and using equation 10, the CDPV limit in a given network can be determined only knowing four parameters of the electricity grid: annual energy and maximum annual power, both for demand and for PV generation. This fact may be of interest when analysing networks that have the same LCD and ODCPV profiles. If these curves are not available, the method shown in figure 2 will have to be applied, i.e., firstly it should be obtained LCD and ODCPV curves for that network should be obtained.” Please, see lines 451-457 (manuscript without changes marked).
Poin 4: So, yes, you identified the maximum theoretical coverage, but what's the significance if that is not feasibile?
Response 4: This is a quite interesting question. In order to clarify this issue, we have modified different parts of the manuscript. Among others, please see Conclussions, lines 481-490 (manuscript without changes marked).
“The limit of the coverage of electricity demand for a given electricity generation technology is an indicator of the potential of this technology and shows the ability of PVs to cover the electricity demand for a given grid. It can be very useful in energy planning of large electricity grids as it provides certainty to future projects of photovoltaic installations about the capacity of the demand to absorb the photovoltaic energy that can be generated. It can be very useful in energy planning of large electricity grids as it provides certainty to future projects of photovoltaic installations about the capacity of the demand to absorb the photovoltaic energy that can be generated. It must be stated that the limit developed in the manuscript provides the potential of PVs alone when facing the demand. Values beyond this limit are not possible if no electrical storage and photovoltaic overgeneration are considered.”
Poin 5: Also, the assumption of no storage and no exchange, how is that realistic?
Response 5: It is realistic for determining this limit. It must be stated that this limit provides the potential of PVs alone when facing the demand. So, neither storage nor inter-network exchanges are considered. As it has been included in the new version of the manuscript (please see lines 94-103, manuscript without changes marked): “Currently, although the level of CDPV is low, there are very significant growth estimations. In the countries with the highest coverage it was slightly higher than 8% while in EU and the world as an average it reached 5% and 3%, respectively, in 2019 [12]. However, this coverage will increase significantly as an annual growth of the new photovoltaic power installed of 9% is expected for the period 2020-2024. [13]. If this growth is kept in the next decades, there will be many countries with a considerable increase in CDPV where PVs may become one of the main sources of electricity generation Nowadays, in most networks, the installed photovoltaic power is still far from reaching CDPVmax. However, the accumulated PV power is continuously increasing and getting closer to this limit. The CDPV limit may provide, given the aforementioned boundary conditions, the maximum photovoltaic power that a given network can absorb without storage and inter-network exchanges”. In this sense, this limit may help to provide not only a proper energy planning of a territory but to analyze the viability of the integration of future photovoltaic projects in the electrical system.

Reviewer 2 Report

General Comments

1.The paper does provide a useful technique that can give a first order approximation of the capacity of PV to meet the demands of a particular electrical grid. However, the usefulness of the technique needs to be demonstrated for the examples of the six grids. The authors should give some more information on those grids to demonstrate the usefulness and limitations of this technique. If this would take too much then the authors can focus on just one case as an example.

The manuscript in lines 501-505 ends with the conclusion that this method is of limited use as it is used only to estimate a theoretical limit. It would be useful to understand what they have right now for those grids (or at least analyse one) and how this technique can help decision making in the future. The manuscript ends on a weak note (lines 501-505) and it needs to be updated to give readers the motivation to use the techniques presented in the manuscript.

  1. There are many sentences that are hard to read. I would recommend a thorough review of the manuscript. In the specific comments I am giving some examples for sentences to be rewritten. Not all of them are given here therefore I suggest a thorough review and rewriting for clarity.

Specific comments

Lines 12-13. The sentence is awkward. Maybe rewrite “In a large electricity grid without storage, there is a limit of the maximum photovoltaic energy that can be consumed as the demand and generation may not match, either in magnitude or in time”.

Lines 17-18. The sentence is awkward. Maybe rewrite “We will assume there is no energy storage or inter-network exchanges”.

Lines 19-20: The sentence “it has been not considered photovoltaic 19 overgeneration and a complete electrical system flexibility” is not clear. Please rewrite.

Line 30: The word simbology maybe should be changed to “Symbology” or “Symbol Definitions”

Lines 58-59. The sentence is unclear. Maybe rewrite to “The daily output power curve of a photovoltaic (PV) power generation plant, in the absence of 58 storage, follows the incident solar radiation”.

Lines 61-62. The explanation for Figure 1 is not clear (what days are chosen and why those are chosen). Also on the text for Figure 1. Also it is not clear how that figure helps the manuscript. Maybe remove it as it does not add much to the content.

Lines 252-254. Sentences like that are obvious and unnecessary. I would suggest to remove them. “This is only due to human beings customs, to 252 which, obviously, has nothing to do with the PV generation, which is governed by incident solar 253 radiation on the PV generator, a fact that makes it difficult to comply with this condition”.

Lines 483-484. The authors claim that all you need is 4 parameters. If you just know the annual energy and maximum annual power, both for demand and for PV generation then you can calculate the limit. But this is not what is shown in this manuscript. The authors do not just start from those 4 parameters then calculate the CDPV limit. The authors use the methodology shown in Figure 4 which is straightforward but requires work and it is not just 4 parameters.

Author Response

The authors would like to thank you for your revision. We kindly appreciate your comments that have helped to improve the manuscript and to make the text clearer.

Point 1: The paper does provide a useful technique that can give a first order approximation of the capacity of PV to meet the demands of a particular electrical grid. However, the usefulness of the technique needs to be demonstrated for the examples of the six grids. The authors should give some more information on those grids to demonstrate the usefulness and limitations of this technique. If this would take too much then the authors can focus on just one case as an example.

Response 1: You are right and it is a very interesting suggestion, but, actually, we don´t have enough time to pursue this issue. However, we are planning a manuscript related with this issue. As you suggest, we will analyze the six grids in order to show the usefulness and limitations of the method developed in the manuscript.

Point 2: The manuscript in lines 501-505 ends with the conclusion that this method is of limited use as it is used only to estimate a theoretical limit. It would be useful to understand what they have right now for those grids (or at least analyse one) and how this technique can help decision making in the future. The manuscript ends on a weak note (lines 501-505) and it needs to be updated to give readers the motivation to use the techniques presented in the manuscript. 

Response 2: You are right. Conclussions have been modified according to your suggestions. Please, see lines 467-489:

“In this paper a new method has been developed to estimate the limit of the CDPV in large electricity grids. Solar PV has a limitation when considering the amount of energy that can be absorbed by a large conventional electrical network. This is due to the matching between the demand curve and the photovoltaic generation curves. Currently, in most networks, the installed photovoltaic power is far from reaching this limit. However, the accumulated PV power is continuously increasing and may get closer to this limit in the next decades. The CDPVMAX may provide, given the aforementioned boundary conditions, the maximum photovoltaic power that a given network can absorb if neither storage nor inter-network exchanges are considered.

The method has the advantage of eliminating the random and the periodic variability over time of the PV power and, to a lesser extent, the demand power. This is achieved through the ODCPV and the LDC curves which, together with the best scenario conditions to maximize the CDPV, make it possible to obtain the theoretical limit which the PV generation can achieve for a given electricity grid. This limit depends both on the profiles of LDC and ODCPV, both stable over time for an electricity grid.

The CDPVmax for a given electricity generation technology is an indicator of the potential of this technology and shows the ability of PVs to cover the electricity demand for a given grid. It can be very useful in energy planning of large electricity grids as it provides certainty to future projects of photovoltaic installations about the capacity of the demand to absorb the photovoltaic energy that can be generated. It can be very useful in energy planning of large electricity grids as it provides certainty to future projects of photovoltaic installations about the capacity of the demand to absorb the photovoltaic energy that can be generated. It must be stated that the limit developed in the manuscript provides the potential of PVs alone when facing the demand. Values beyond this limit are not possible if no electrical storage and photovoltaic overgeneration are considered”

Point 3: There are many sentences that are hard to read. I would recommend a thorough review of the manuscript. In the specific comments I am giving some examples for sentences to be rewritten. Not all of them are given here therefore I suggest a thorough review and rewriting for clarity.

Response 3: We have tried to clarify the manuscript and, as you suggest, a deep review has been made. Moreover, it has been revised by a native speaker.

Specific comments

All the specific comments have included in the manuscript as you suggested. Thanks for your time and interest.

Point 4: Lines 12-13. The sentence is awkward. Maybe rewrite “In a large electricity grid without storage, there is a limit of the maximum photovoltaic energy that can be consumed as the demand and generation may not match, either in magnitude or in time”.

Response 4: Please, see lines 11-13

Point 5: Lines 17-18. The sentence is awkward. Maybe rewrite “We will assume there is no energy storage or inter-network exchanges.

Response 5: Please, see lines 16-17.

Point 6: Lines 19-20: The sentence “it has been not considered photovoltaic overgeneration and a complete electrical system flexibility” is not clear. Please rewrite.

Response 6: Please, see lines 18-19.

Point 7: Line 30: The word simbology maybe should be changed to “Symbology” or “Symbol Definitions”.

Response 7: Please, see line 513. Symbology has moved to Abbreviations Section.

Point 8: Lines 58-59. The sentence is unclear. Maybe rewrite to “The daily output power curve of a photovoltaic (PV) power generation plant, in the absence of storage, follows the incident solar radiation”.

Response 8: Please, see lines 40-41(manuscript without changes marked).

Point 9: Lines 61-62. The explanation for Figure 1 is not clear (what days are chosen and why those are chosen). Also on the text for Figure 1. Also it is not clear how that figure helps the manuscript. Maybe remove it as it does not add much to the content.

Response 9: You are right. As you suggest, the figure and the associated text have been removed. Moreover, figure 2 has also been removed. Load generation and demand curves including matching can be clearly seen in figure 3.

Point 10: Lines 252-254. Sentences like that are obvious and unnecessary. I would suggest to remove them. “This is only due to human beings customs, to 252 which, obviously, has nothing to do with the PV generation, which is governed by incident solar 253 radiation on the PV generator, a fact that makes it difficult to comply with this condition”.

Response 10: Thanks for your suggestion, they have been removed from the text.

Point 11: Lines 483-484. The authors claim that all you need is 4 parameters. If you just know the annual energy and maximum annual power, both for demand and for PV generation then you can calculate the limit. But this is not what is shown in this manuscript. The authors do not just start from those 4 parameters then calculate the CDPV limit. The authors use the methodology shown in Figure 4 which is straightforward but requires work and it is not just 4 parameters.

Response 11: The paragraph “The method here developed manages to determine the limit in the CDPV where only four parameters of the electricity grid are needed: annual energy and maximum annual power, both for demand and for PV generation.” is replaced by “Through this method, if LCD and OCDPV curves are known, the aforementioned boundary conditions are set and using equation 10, the CDPV limit in a given network can be determined only knowing four parameters of the electricity grid: annual energy and maximum annual power, both for demand and for PV generation. This fact may be of interest when analysing networks that have the same LCD and ODCPV profiles. If these curves are not available, the method shown in figure 2 will have to be applied, i.e., firstly it should be obtained LCD and ODCPV curves for that network should be obtained.”  Please, see lines 451-457.

Reviewer 3 Report

The topic is interesting and it is adapt to this journal. The collaboration among several faculties is useful and I think that there is a great work behind the presentation of this work. However, while the presentation is nice in shape, there are few comments and/or suggestions to improve the manuscript.

-According to scientific standards, abbreviations cannot be used in the abstract, please correct it in the manuscript.

-Clarify better the innovation of this work in the abstract and in the main text.

-Read articles to understand the structure of Sensors. The following structure would be preferable based on the Sensors Microsoft Word template file: 1. Introduction (1.1, 1.2, 1.3.), 2. Materials and Methods (2.1, 2.2., 2.3.), 3. Results and Discussion (3.1, 3.2, 3.3), 4. Conclusions. These sections mixed in the text. The introduction section is a literary review of the topic. In the introduction (or where still necessary) all paragraph must be cited because of the risk of plagiarism.

https://www.mdpi.com/journal/sensors/instructions

-The following parts are missing after the conclusion: Author Contributions; Acknowledgments; Conflicts of Interest; Abbreviations

-The "SIMBOLOGY" and "ACRONYMS" should be moved to the abbreviations section.

-The citation style is inappropriate. Mendeley can easily solve this problem. In Mendeley can be found also the style of Energies (Mendeley is a free reference manager and an academic social network): https://www.mendeley.com/ 

-Please provide more general and global information to the introduction because this part is incomplete. It would be necessary to mention more the problem of variable renewable energies (VRE) and electricity balancing in the introduction. Growing share of intermittent renewable electricity generation, as well as changing patterns in electricity demand create challenges for not only grid balancing but security of supply, too. It would be necessary to mention more this problem in the introduction. For example, this manuscript can provide good general information. https://www.mdpi.com/2071-1050/12/3/811

-In addition, it would be important to compare the results with other „new” modeling concepts:

http://tool.european-calculator.eu/intro

http://www.european-calculator.eu/documentation/

https://tyndp.entsoe.eu/tyndp2018/scenario-report

-Please search references to the equations. Equations should always be accurately and clearly referenced.

-Please add more information's about the model validation in a new chapter.

-Extend the conclusion with more general usability. What are the benefits of the results in a global context? Please explain this better in the manuscript.

-At the end of the study need to create a nomenclature table (abbreviations section) with units.

Author Response

The authors would like to thank you for your revision. We kindly appreciate your comments that have helped to improve the manuscript and to make the text clearer.

Point 1: According to scientific standards, abbreviations cannot be used in the abstract, please correct it in the manuscript.

Response 1: You are right, abbreviations should not be used in the abstract. They have been removed. Please, see abstract.

Point 2: Clarify better the innovation of this work in the abstract and in the main text.

Response 2: The following paragraph has been included in Introduction. Please see lines 80-88.

“Solar PV has a limitation when considering the amount of energy that can be absorbed by a large conventional electrical network. This is due to the matching between/of the demand curve and the photovoltaic generation curves. Currently, in most networks, the installed photovoltaic power is far from reaching this limit. However, the accumulated PV power is continuously increasing and getting closer to this limit. The CDPV limit may provide, given the aforementioned boundary conditions, the maximum photovoltaic power that a given network can absorb.”  

Point 3: Read articles to understand the structure of Sensors. The following structure would be preferable based on the Sensors Microsoft Word template file: 1. Introduction (1.1, 1.2, 1.3.), 2. Materials and Methods (2.1, 2.2., 2.3.), 3. Results and Discussion (3.1, 3.2, 3.3), 4. Conclusions. These sections mixed in the text.

Response 3: You are right. As you suggest, we have restructured the manuscript according to the template. Moreover, Section 3.1 Data preprocessing has moved to section 2 Materials and methods. Please see manuscript.

Point 4: The introduction section is a literary review of the topic. In the introduction (or where still necessary) all paragraph must be cited because of the risk of plagiarism.

https://www.mdpi.com/journal/sensors/instructions

Response 4: It has been revised again and it has been cited all the needed references.

Point 5: The following parts are missing after the conclusion: Author Contributions; Acknowledgments; Conflicts of Interest; Abbreviations.

Response 5: You are right. The aforementioned parts have been included in the manuscript. Please, see lines 504-530

Point 6: The "SIMBOLOGY" and "ACRONYMS" should be moved to the abbreviations section.

Response 6: "SIMBOLOGY" and "ACRONYMS" have been moved to Abbreviations section.

Point 7: The citation style is inappropriate. Mendeley can easily solve this problem. In Mendeley can be found also the style of Energies (Mendeley is a free reference manager and an academic social network): https://www.mendeley.com/

Response 7: You are right, we have modified the references following the template indications. Please, see References.

Point 8: Please provide more general and global information to the introduction because this part is incomplete. It would be necessary to mention more the problem of variable renewable energies (VRE) and electricity balancing in the introduction. Growing share of intermittent renewable electricity generation, as well as changing patterns in electricity demand create challenges for not only grid balancing but security of supply, too. It would be necessary to mention more this problem in the introduction. For example, this manuscript can provide good general information. https://www.mdpi.com/2071-1050/12/3/811

Response 8:

Please see lines 33-44.

“From a balancing perspective, the intermittent nature of these renewable energy sources are creating challenges in the electricity sector that previously were not of concern [3].”

and lines 94-103 (manuscript without changes marked).

“Currently, although the level of CDPV is low, there are very significant growth estimations. In the countries with the highest coverage it was slightly higher than 8% while in EU and the world as an average it reached 5% and 3%, respectively, in 2019 [12]. However, this coverage will increase significantly as an annual growth of the new photovoltaic power installed of 9% is expected for the period 2020-2024. [13]. If this growth is kept in the next decades, there will be many countries with a considerable increase in CDPV where PVs may become one of the main sources of electricity generation Nowadays, in most networks, the installed photovoltaic power is still far from reaching CDPVmax. However, the accumulated PV power is continuously increasing and getting closer to this limit. The CDPV limit may provide, given the aforementioned boundary conditions, the maximum photovoltaic power that a given network can absorb without storage and inter-network exchanges.”

[3] Miklós Gyalai-Korpos et al. The Role of Electricity Balancing and Storage: Developing Input Parameters for the European. Calculator for Concept Modeling. Sustainability 2020, 12, 811

[12] IEA PVPS. Snapshot of global photovoltaic Markets 2020. Report IEA PVPS T1-37:2020. April 2020.

[13] Solar Power Europe. Global Market Outlook For Solar Power / 2020-2024. June 2020.

Point 9: In addition, it would be important to compare the results with other „new” modeling concepts:

http://tool.european-calculator.eu/intro13

http://www.european-calculator.eu/documentation/

https://tyndp.entsoe.eu/tyndp2018/scenario-report

Response 9: You are right and it is a very interesting suggestion, but, actually, we don´t have enough time to pursue this issue. However, as you suggest we will compare the results with these new modeling concepts.

Point 10: Please add more information's about the model validation in a new chapter.

Response 10: You are right and it is a very interesting suggestion, but, actually, we don´t have enough time to pursue this issue. However, we are planning a manuscript where it will be analyzed the six grids in order to show the usefulness and limitations of the method developed in the manuscript.

Point 11: Please search references to the equations. Equations should always be accurately and clearly referenced.

Response 11: Most of equations have been developed by authors. However, it has been referenced those ones that can be found in the literature.

Point 12: Extend the conclusion with more general usability. What are the benefits of the results in a global context? Please explain this better in the manuscript.

Response 12: As you suggest, the following paragraphs have been included in Conclussions in order to clarify and emphasize the new method.

Lines 467-474.

“In this paper a new method has been developed to estimate the limit of the CDPV in large electricity grids. Solar PV has a limitation when considering the amount of energy that can be absorbed by a large conventional electrical network. This is due to the matching between the demand curve and the photovoltaic generation curves. Currently, in most networks, the installed photovoltaic power is far from reaching this limit. However, the accumulated PV power is continuously increasing and may get closer to this limit in the next decades. The CDPVMAX may provide, given the aforementioned boundary conditions, the maximum photovoltaic power that a given network can absorb if neither storage nor inter-network exchanges are considered “

Lines 481-489.

“The CDPVMAX for a given electricity generation technology is an indicator of the potential of this technology and shows the ability of PVs to cover the electricity demand for a given grid. It can be very useful in energy planning of large electricity grids as it provides certainty to future projects of photovoltaic installations about the capacity of the demand to absorb the photovoltaic energy that can be generated. It can be very useful in energy planning of large electricity grids as it provides certainty to future projects of photovoltaic installations about the capacity of the demand to absorb the photovoltaic energy that can be generated. It must be stated that the limit developed in the manuscript provides the potential of PVs alone when facing the demand. Values beyond this limit are not possible if no electrical storage and photovoltaic overgeneration are considered.”

Point 13: At the end of the study need to create a nomenclature table (abbreviations section) with units.

Response 13: It has been included. Again, thanks so much for your comments and suggestions.

Round 2

Reviewer 1 Report

The authors have implemented a good revision of the structure and of the language.

Overall, the article is of interest.

The Introduction is quite long but it includes also the literature review, so it could be ok as it is or it could be split into two separate sections.

In general, I still find it difficult to read a strong clarification about the theoretical basis of this study: the conceptualized "maximum coverage by PV" would only be real if the peaks of generation did match with the peaks of demand, which cannot be known in advance. So, it is a conceptual limit that aims at consistency between different analyses, rather than at plausible occurrence of that coverage fraction. This should be stressed more in the discussion/conclusions. Also, when the authors say that "usual methods depend on the considered year and network status", they should clarify that the new method has its own limitations.

There is a typo in Ref. 15's title.

If the authors have suggestions on how to go further and include some energy storage (e.g., as a % of the PV power or of the average daily demand or others), they could propose it in the conclusions (and possibly develop it in a future study, which would be of interest).

Author Response

Reviewer 1

The authors are very thankful for your revisions. We kindly appreciate your comments and suggestions that have helped to improve the manuscript and to make the text clearer.

Comments and Suggestions for Authors

The authors have implemented a good revision of the structure and of the language.

Overall, the article is of interest.

Point 1. The Introduction is quite long but it includes also the literature review, so it could be ok as it is or it could be split into two separate sections.

Response 1. As you suggest, the Introduction has been split into two sections:

1.1. Maximum coverage of the demand by photovoltaics. Definition.
1.2. Methods to estimate the upper limit of the CDPV in a large electricity grid. State of the art

Point 2. In general, I still find it difficult to read a strong clarification about the theoretical basis of this study: the conceptualized "maximum coverage by PV" would only be real if the peaks of generation did match with the peaks of demand, which cannot be known in advance. So, it is a conceptual limit that aims at consistency between different analyses, rather than at plausible occurrence of that coverage fraction. This should be stressed more in the discussion/conclusions. Also, when the authors say that "usual methods depend on the considered year and network status", they should clarify that the new method has its own limitations.

Response 2. We really appreciate your time and effort to improve our manuscript and make the text clearer. We have done our best when explaining this concept as we know that it can be a little bit confusing. In this sense, in the previous version of the manuscript it has been including the following paragraphs regarding the maximum coverage of the demand by photovoltaics (CDPVmax).

Lines 69-74
“Without going into other considerations and only taking into account the behavior of the demand and photovoltaic generation curves, in a traditional power grid, it can be said that the matching between them will never be 100%, without considering any kind of electricity storage. This fact will imply that there will be a limit on the maximum coverage of the demand by photovoltaics (CDPVmax). This parameter, or average penetration, is the proportion of electric energy which is being supplied from photovoltaic plants. The coverage of demand is defined by the following equation [5]:”

Lines 78-83
“The limit of the coverage of electricity demand for a given electricity generation technology is an indicator of the potential of this technology. Therefore, the limit of the CDPV (CDPVmax) is an indicator of the ability of photovoltaic solar energy to cover the electricity demand for a given grid. Solar PV has a limitation when considering the amount of energy that can be absorbed by a large conventional electrical grid. This is due to the matching between the demand curve and the photovoltaic generation curves”

Moreover, as you suggest, we have modified the following paragraph trying to explain better this concept. Please, see lines 484-489.

“The CDPVMAX constitutes a theoretical limit that shows the potential of this technology and the ability of PVs to cover the electricity demand for a given grid. Values beyond this limit are not possible if no electrical storage and photovoltaic overgeneration are considered. For this reason, to know this limit can be very useful in energy planning of large electricity grids as it provides certainty to future projects of photovoltaic installations about the capacity of the demand to absorb the photovoltaic energy that can be generated.”

It must be emphasized that knowledge of the limitations of a technology is of great importance to make a realistic and effective use of it. As in the solar cell based on a single PN junction, the efficiency limit was determined to be around 33.7% for a prohibited band of 1.34 eV (Shockley-Queisser), indicating that in these boundary conditions any attempt to overcome this efficiency is not possible. Please excuse our vanity for setting such an example, but we think that it may help to understand the meaning of CDPVmax.

William Shockley and Hans J. Queisser, "Detailed Balance Limit of Efficiency of p-n Junction Solar Cells", Journal of Applied Physics 1961, 32, 510-519; doi 10.1063/1.1736034.

Point 3. There is a typo in Ref. 15's title.

Response 3. Thanks again for your time and interest. It has been corrected.

Point 4. If the authors have suggestions on how to go further and include some energy storage (e.g., as a % of the PV power or of the average daily demand or others), they could propose it in the conclusions (and possibly develop it in a future study, which would be of interest).

Response 4. You are right. After the paragraph “In addition, this limit can be increased by means of improvements in PV generation, such as the incorporation of the geographic dispersion between PV plants, solar tracking, PV overcapacity and the storage of photovoltaic generation surplus. Furthermore, by managing the electricity demand, this limit can be also increased this limit, e.g.: if the peak of daily demand shifts towards solar noon and the peak of annual demand towards the summer solstice, the valley of daily demand during the night or increases demand on weekends and holidays.” We have included in Conclusions the following paragraph (please see lines 503-505):

“Future works should be developed in this sense in order to know the effect of the aforementioned factors when determining the limit on the coverage of the demand by photovoltaics.”

Reviewer 2 Report

The manuscript is improved and the issues raised have been adequately addressed. I recommend another careful review to ensure that no errors remain in the final manuscript.

Lines 18-19.  Saying that system flexibility have not been considered is not correct. I think you meant to say "photovoltaic overgeneration is not considered and a complete system flexibility is assumed".

Line 85. There is a missing ".".  It should be " ... absorb. In this ..."

Line 153: "Th values..." should be "The values..."

Lines 498. The word "limit" appears twice. Rewrite that sentence.

Author Response

Reviewer 2

The authors are very thankful for your revisions. We kindly appreciate your comments and suggestions that have helped to improve the manuscript and to make the text clearer.

Comments and Suggestions for Authors

The manuscript is improved and the issues raised have been adequately addressed. I recommend another careful review to ensure that no errors remain in the final manuscript.

Point 1. Lines 18-19. Saying that system flexibility have not been considered is not correct. I think you meant to say "photovoltaic overgeneration is not considered and a complete system flexibility is assumed".

Response 1. You are right. It has been modified following your suggestion. Thanks again for your interest.

Point 2. Line 85. There is a missing ".". It should be " ... absorb. In this ...".

Response 2. It has been included (please see line 85).

Point 3. Line 153: "Th values..." should be "The values...".

Response 3. It has been corrected (please see line 155).

Point 4. Lines 498. The word "limit" appears twice. Rewrite that sentence.

Response 4. Sorry for the typographic errors. It has been eliminated. (Please see line 501).
